# Mitochondria and Doxorubicin-Induced Cardiomyopathy: A Complex Interplay

**DOI:** 10.3390/cells11132000

**Published:** 2022-06-22

**Authors:** Leonardo Schirone, Luca D’Ambrosio, Maurizio Forte, Riccardo Genovese, Sonia Schiavon, Giulia Spinosa, Giuliano Iacovone, Valentina Valenti, Giacomo Frati, Sebastiano Sciarretta

**Affiliations:** 1Department of Medical-Surgical Sciences and Biotechnologies, Sapienza University of Rome, Corso della Repubblica 79, 04100 Latina, Italy; leonardo.schirone@uniroma1.it (L.S.); l.dambrosio@uniroma1.it (L.D.); riccardogenovese1996@gmail.com (R.G.); sonia.schiavon@uniroma1.it (S.S.); giu.spi15@gmail.com (G.S.); giacomo.frati@uniroma1.it (G.F.); 2Department of Angiocardioneurology, IRCCS Neuromed, 86077 Pozzilli, Italy; maurizio.forte@neuromed.it; 3Cardiology Division, Santa Maria Goretti Hospital, 04100 Latina, Italy; giuliano.iacovone@uniroma1.it (G.I.); valevale2012@hotmail.com (V.V.)

**Keywords:** DOX, anthracycline, cardiomyopathy, mitochondria, heart

## Abstract

Cardiotoxicity has emerged as a major side effect of doxorubicin (DOX) treatment, affecting nearly 30% of patients within 5 years after chemotherapy. Heart failure is the first non-cancer cause of death in DOX-treated patients. Although many different molecular mechanisms explaining the cardiac derangements induced by DOX were identified in past decades, the translation to clinical practice has remained elusive to date. This review examines the current understanding of DOX-induced cardiomyopathy (DCM) with a focus on mitochondria, which were increasingly proven to be crucial determinants of DOX-induced cytotoxicity. We discuss DCM pathophysiology and epidemiology and DOX-induced detrimental effects on mitochondrial function, dynamics, biogenesis, and autophagy. Lastly, we review the current perspectives to contrast the development of DCM, which is still a relatively diffused, invalidating, and life-threatening condition for cancer survivors.

## 1. Doxorubicin-Induced Cardiomyopathy: Pathophysiology and Epidemiology

Doxorubicin is an effective anthracycline used as a chemotherapeutic agent in several malignancies. However, its use may be limited by severe adverse effects that have been reported over the past decades, notably cardiotoxicity [1]. Doxorubicin-induced cardiomyopathy (DCM) is a potentially lethal condition that fails to benefit from available therapies and may manifest acutely or chronically [2]. The acute form usually emerges within 2–3 days from the chemotherapeutic treatment and is characterized by tachycardia, electrocardiographic abnormalities, premature beats, myopericarditis, and, sometimes, acute left ventricular failure [3]. Myocardial edema, a treatable and reversible condition, is the more likely pathophysiological condition responsible for acute DCM, although the mechanisms for these alterations still need to be clarified [2]. Acute DCM has an incidence of approximately 11%, whereas chronic DCM is calculated to be close to 2% [2]. The significant and variable time gap between treatment and chronic DCM manifestation ranges from 30 days to more than 10 years and represents a challenge for accurate characterization of DCM epidemiology [4]. Young and advanced age and a history of cardiovascular disease are risk factors for the terminal evolution of chronic DCM to congestive heart failure (CHF). This severe life-threatening cardiac dysfunction kills 50% of the affected patients within 1 year [4]. The received lifetime dose impacts the incidence, which ranges from 5% (400 mg/m^2^) to 48% (over 700 mg/m^2^), and the European Society of Cardiology provided guidelines in 2016 to manage anthracycline-induced cardiac complications [5]. A retrospective pooled analysis of trials delivered the current basis for setting the lifetime cumulative dose to a maximum of 450 mg/m^2^ [6]. In addition, subclinical cardiotoxicity without overt cardiomyopathy development is characterized by an increase in cardiac biomarkers (i.e., BNP and troponin) in 30–35% of treated cancer patients, with arrhythmias in more than 12% of patients, and subtle changes of cardiac structure and function (9–11%) [1]. Sex-related differences in response to DOX were also observed with different results. Female sex was found to be a significant risk factor for DCM in pediatric cancer patients [7,8]. A high risk of developing CHF in adult females receiving DOX was also reported [9]. However, other studies showed that the female sex was protective for left ventricular systolic dysfunction development in response to anthracycline treatment [10]. Male sex was reported to be a risk factor for symptomatic heart failure or cardiac death [11]. Other reports failed to find sex-related differences in response to DOX or other anthracycline treatments [12,13]. One of the possible explanations for the divergent results observed in adult patients is that many studies about the effects of DOX were conducted in breast cancer patients, predominantly composed of females. Thus, the ratio between males and females for an appropriate comparison appears unbalanced in most cases. In addition, estrogens might account for the cardiac protection in females that was reported in some studies. Of course, these differences may also depend on the drug used and its dose.

Extremes of age were also reported to be essential risk factors for anthracycline-induced cardiotoxicity (age >65 years or <4 years) [14]. Older age correlates with a higher risk of developing cardiomyopathy in patients with lymphoma [15]. Cardiotoxicity in the elderly manifests earlier, more frequently in case of coexisting cardiac risk factors, such as hypertension. Notably, premature signs of cardiac aging were documented in cancer survivors treated with DOX [16]. Other observations suggest that patients <4 years and those >65 years showed an increased risk of developing DCM, likely due to the reduced DOX clearance by the liver [17].

Nearly 80% of DCM patients suffer from initial symptoms of heart failure, including orthopnoea, excessive sweating, ankle edema, and fatiguability. In advanced cases, the clinical examination of these patients often reveals tachycardia, elevated jugular pressure, pulmonary crepitations, and peripheral or sacral edema [18]. Cardiac preload and afterload are higher than usual, resulting in elevated wall stress impairing diastolic filling, often coupled with a reduction in systolic function [18]. This is often linked with a maladaptive fibrotic reparative response in the midwall affecting one-third of patients with advanced DCM, which is associated with increased cardiomyocyte death [19]. Moreover, the neurohormonal system is activated, increasing adrenergic activity, stimulating the renin-angiotensin-aldosterone (ACE) system, and the release of natriuretic peptides, e.g., atrial natriuretic peptide (ANP) [20,21,22]. These combined effects result in increased cardiac afterload that initially leads to compensative hypertrophy and ultimately to left ventricle (LV) remodeling [23]. Right ventricular remodeling may also be observed in the late phases of DCM due to extended damage to the myocardium or secondary pulmonary hypertension [23].

DCM severity encouraged scientists to investigate the molecular mechanisms by which DOX exerts cytotoxic effects in terminally differentiated cells (i.e., cardiomyocytes) and to develop a sustainable cardioprotective strategy [24]. Several mechanisms have been proposed in past years (Figure 1) and were extensively discussed in previous comprehensive reviews [25,26,27]. In this review, we focus on the prominent role of mitochondria in mediating the detrimental cardiac side-effects of DOX.

## 2. DOX-Induced Mitochondrial Dysfunction

### 2.1. Cell Death

Mitochondrial dysfunction was the first observed mechanism associated with cardiomyocyte toxicity induced by DOX. In fact, Doroshow and Davies reported in the ’80s that DOX generates superoxide anion and ROS in bovine submitochondrial heart preparations, which in turn contribute to the derangement of the mitochondrial electron transport chain (ETC), particularly complex I [28,29]. The specificity of DOX for the heart was explained by its high affinity for cardiolipin, a phospholipid particularly abundant in the inner mitochondrial membrane (IMM), and by the elevated mitochondrial mass that characterizes the heart to meet its energy demand [30]. DOX irreversibly binds cardiolipin, impairing mitochondrial membrane integrity and subtracting a crucial anchor point for cytochrome C [31]. The latter triggers a harmful radical chain reaction that culminates in apoptosis [32]. The importance of this interaction was also confirmed by treating with DOX cardiolipin-deficient human lymphocytes harvested from patients suffering from Barth’s syndrome, a multiorgan genetic disease originating from the mutation of Tafazzin (TAZ), an essential protein for cardiolipin biosynthesis. These cells display resistance to DOX-induced damage compared to those derived from healthy patients [33]. In addition, H9C2 cardiac cells silenced for TAZ show reduced DOX-induced oxidative damage [33]. These results suggest that cardiolipin represents an important target in DOX-induced cardiotoxicity.

Despite being generally associated with a protective cardiovascular effect, endothelial nitric oxide synthase (eNOS) also appears to play an essential role in mediating the acute cardiotoxic effects of DOX. eNOS-KO mice treated with a single dose of DOX (20 mg/kg) show preserved cardiac function and reduced cell death compared to wild-type and transgenic eNOS overexpressing animals [34]. However, mice receiving fenofibrate display preserved cardiac function and structure through the activation of the Akt/eNOS pathway in response to chronic administration of the same cumulative dose of DOX (20 mg/kg), whereas pharmacological eNOS inhibition through L-NAME abolishes the protective effects of fenofibrate [35]. These studies prove that acute and chronic DCM may be characterized by differential molecular signatures underlying the development of cardiac dysfunction.

Besides cardiolipin, increased cardiomyocyte ROS production was also associated with mitochondrial iron overload in human heart specimens from patients suffering from DCM but not from other cardiomyopathies [36]. Of note, cardiac-specific overexpression of ABCB8, a mitochondrial exporter of iron, markedly reduces ROS production and myocardial dysfunction in a murine model of DCM [36]. Marked iron accumulation eventually leads to cardiomyocyte ferroptosis, a peculiar form of non-apoptotic programmed cell death involving lipid peroxidation [37]. In several independent studies, ferroptotic cell death was countered by administering dexrazoxane, which improves redox status and preserves myocardial function in murine preclinical models of DOX treatment using mice and rats [36,38,39]. Dexrazoxane is FDA-approved for the cardioprotection of oncological patients and was successfully used in pediatric patients with acute lymphoblastic leukemia [40] and, to date, is being tested in breast cancer patients in the PHOENIX1 phase I clinical trial (NCT03930680).

Together with ferroptosis, DOX accumulation may lead to a high-conductance mitochondrial permeability transition pore (mPTP) opening [41], a phenomenon that releases mitochondrial Ca^2+^ and Cytochrome C contributing to apoptotic or necrotic cell death, depending on the nature of the upstream stress [42,43]. Cyclosporin-A (CyA), an mPTP opening inhibitor, was administered to mice chronically receiving DOX and was proven to protect myocardial function and reduce cardiomyocyte mortality after 16 weeks [44]. Mitochondrial-dependent necrotic cell death was found to be linked to DOX-induced repression of nuclear factor kappa-light-chain-enhancer of activated B cells (NF-κB), which leads to increased levels of the mitochondrial B-cell lymphoma 2 (BCL-2)-Interacting Protein 3 (BNIP-3) [45,46]. Systemic BNIP-3^−/−^ mice treated with DOX show normal heart function, absence of myocardial remodeling, and preserved mitochondrial respiration, confirming a prominent role for this protein in DCM [47]. The cardiotoxic induction of BNIP-3 in response to DOX was also reduced in mice overexpressing translationally controlled tumor protein (TCTP), which is downregulated in response to DOX, thereby triggering cardiomyocyte death and heart dysfunction [48]. In addition, chronic DOX-induced contractile dysfunction and cardiomyocyte apoptosis are reduced in p53^+/−^ and in mice with cardiac overexpression of BCL-2 by impairing the oxidative DNA damage/Ataxia telangiectasia mutated (ATM)/p53 pro-apoptotic pathway [49].

Together, these studies indicate that inhibiting pro-apoptotic pathways protects the heart from developing DCM in animal models.

### 2.2. Metabolic Derangements

Besides cell death mechanisms, also mitochondrial metabolism is impaired in DOX-induced cardiotoxicity. In metabolomic studies conducted on rats, disorders with energy metabolism, fatty acids oxidation, amino acids, purine and choline metabolism, and gut microbiota-related metabolism were all associated with DOX administration [50]. Treated rats display a reduction in crucial mitochondrial elements, including superoxide dismutase (SOD) and nuclear respiratory factor 2 (NRF-2), which mediate stress response, and protein tyrosine phosphatase 1B (PTP1B), whose activation impairs insulin receptor substrate 1 (IRS-1), leading to disorders of fatty acid metabolism and glycolysis [51]. Aldose reductase promotes high glucose-induced superoxide overproduction insult. Its inhibition with fidarestat preserved heart function, reduced circulating troponin I, and increased mitochondrial biogenesis in nude mice receiving colorectal cancer xenografts and treated with DOX [52,53]. This effect might be linked to the activation of the Sirtuin 1—Peroxisome proliferator-activated receptor (PPAR)γ coactivator 1-alpha (SIRT1-PGC-1α)/NRF2 pathway, similar to the effects of aldose reductase inhibition in models of diabetic cardiomyopathy. Future studies testing this hypothesis in DCM models are warranted [54].

Energy metabolism derangements after DOX treatment were associated with a cardiac downregulation of PPAR-α [55], PPAR-γ [56], and PPAR-δ [57], which are transcription factors involved in the regulation of genes regulating carbohydrate, lipid, and protein metabolism. Administration of pharmacological agonists of PPARs was found to preserve heart function and reduce damage markers in murine models of DOX administration. Tumor-bearing wild-type mice receiving 4 mg/kg DOX every week for 6 weeks (to a final cumulative dose of 24 mg/kg) and fenofibrate (a PPAR-α agonist) display preserved heart function and cell survival compared to DOX-treated control group. This effect is recapitulated by overexpressing PPAR-α through a recombinant adeno-associated virus serotype 9 (rAAV9). The protection from DOX-associated toxicity exerted by PPAR-α activation is nullified by the mesenchyme homeobox 1 (MEOX1) knockdown, demonstrating a primary role for this factor in this model [55]. Mice receiving two weeks of pre-treatment with piperine, an agonist of PPAR-γ, and acute DOX treatment at 15 mg/kg display reduced cardiac dysfunction, ROS production, and cell death. These effects are lost if mice receive a pharmacological antagonist of PPAR-γ [56]. Similarly, Wistar rats receiving 15 mg/kg DOX and the PPAR-δ agonist GW0742 display preserved contractile function and reduced troponin phosphorylation compared to those receiving only DOX [57]. These results suggest that PPAR activation may be suitable for mitigating DOX-induced cardiotoxicity.

Mice receiving the lipid catabolism inducer adiponectin show reduced fibrosis and apoptosis after chronic DOX treatment, an effect lost if DOX is administered together with dorsomorphin, an AMP-activated protein kinase (AMPK) inhibitor [58]. AMPK is a keystone of energy metabolism in peripheral tissues. It responds to hormonal signals (e.g., adiponectin and leptin) to recover the energy status by stimulating glucose and lipid consumption and inhibiting anabolism [59]. In fact, knockdown of the glycolytic protein α-enolase rescues myocardial contraction, mitochondrial dysfunction, apoptosis, and AMPK dephosphorylation in the hearts of rats receiving chronic DOX administration [60]. AMPK activation was also observed in rats receiving oleuropein. This natural phenolic compound shows reduced metabolomic derangements after chronic DOX treatment, preserved cardiac function, reduced markers of inflammatory damage, and iNOS inhibition [61]. These results suggest that DOX induces detrimental cardiac effects by reducing AMPK activity.

Inflammatory balance is also crucial in DCM, as observed in mice receiving the interferon γ (IFNγ) inhibitor ‘R46-A2’ in acute DOX-induced cardiotoxicity that had improved cardiac function compared to those that received only DOX [62]. Of note, mice lacking the critical inflammatory receptor Toll-like receptor 4 (TLR-4) show preserved myocardial function, physiological inflammatory levels, and reduced apoptosis in response to acute DOX administration, demonstrating the relevance of inflammation in mediating the early cardiotoxicity of DOX [63]. Similarly, inflammasome-deficient Nod-like receptor protein 3 (NLRP-3) KO mice have reduced levels of DOX-induced pyroptosis and do not display myocardial structural and functional abnormalities after chronic DOX treatment. This mechanism involves the activation of NADPH oxidase 1 (NOX-1), NOX-4, and Caspase 1 by DOX, as proven by mechanistic experiments with Caspase 1 KO mice and pharmacological NOX inhibition (GKT137831), which abolish cardiotoxicity [64].

Overall, these studies show that preserving energy metabolism is a suitable strategy to mitigate the cardiotoxic effects of DOX (Figure 2).

## 3. DOX-Impaired Mitochondrial Biogenesis and Dynamics

### 3.1. Mitochondrial Biogenesis

Several lines of evidence demonstrated that mitochondrial biogenesis is impaired in the heart of murine models receiving DOX. Increased mitochondrial biogenesis was observed in DOX-treated mice that received ferruginol, a pharmacological activator of PPARγ coactivator 1 alpha (PGC-1α), the master regulator of mitochondrial biogenesis. This was associated with increased fatty acid oxidation, mitochondrial function, and myocardial integrity [65].

Mitochondrial DNA (mtDNA) integrity in the presence of ROS and DNA poisoners is essential for preserving mitochondrial biogenesis since mtDNA retains some of the unique para-prokaryotic features of mitochondrial biology. It is known that DOX forms a ternary complex with the DNA-binding enzyme topoisomerase 2 (TOP-2) and cleaved DNA, causing double-strand breaks in proliferating cells. Ternary complexes were also reported in non-dividing cells, along with dramatic signs of mitochondrial dysfunction after DOX treatment. TOP-2β KO mice are protected from DOX-induced cardiac derangements and display unimpaired mitochondrial biogenesis [66]. Interestingly, TOP-2β can be inhibited by dexrazoxane, further supporting this drug’s cardioprotective efficacy. However, a severe downside is that the inhibition of the topoisomerase-related poisoning effects of DOX in cardiomyocytes might, at the same time, also reduce its antineoplastic efficacy [67]. In fact, patients with mutations or low levels of topoisomerase II are less responsive to chemotherapy [68,69]. This led in 2011 to the restriction of dexrazoxane to breast cancer patients receiving more than 300 mg/m^2^ of DOX, following the idea that it might increase the risk of second primary malignancies and myelosuppression. However, recent risk-benefit analyses validated the safety, cardioprotective efficacy, and chemotherapeutic compatibility associated with the administration of this drug, persuading the Committee for Medicinal Products for Human Use and the European Medical Agency to lift the limitations, contraindicating it only to younger subjects (0–18 years) that received less than 300 mg/m^2^ of DOX or an equivalent chemotherapeutic agent [70].

An opposite effect in DOX-induced cardiotoxicity is exerted by the mitochondrial topoisomerase TOP1mt, which was shown to be a core player in mitochondrial homeostasis and resistance to genotoxic stress. Loss of this mitochondrial enzyme exacerbates DOX-induced cardiotoxicity, indicating that the effects of different members of this enzymatic family are not redundant [71].

Previous work also showed that NRF-2 might affect DOX-induced cardiotoxicity and alterations of mitochondrial biogenesis. Systemic deficiency of NRF-2 was associated with exacerbated cardiac derangements in response to acute DOX injection (25 mg/kg). In vitro, NRF-2 downregulation recapitulates cardiomyocyte damage induced by DOX treatment, while its overexpression strongly reduces DOX cytotoxicity by ROS scavenging and autophagy activation [72]. NRF-2 also controls mitochondrial biogenesis downstream to a pathway involving heme oxygenase (HO)-1 and Akt, which is activated by ROS [73]. Of note, chronic models of DOX administration to rats showed that the Kelch-like ECH-associated protein 1 (KEAP-1)/NRF-2 pathway is inhibited in these animals, accounting for the progressive degeneration of myocardial integrity and function [74].

Other studies demonstrated that DOX impairs the function of fundamental mitochondrial components by acting on genes encoded by both mtDNA and nuclear DNA [75]. Loss-of-function mutations in mtDNA-encoded complex IV subunits were found in the hearts of rats exposed to chronic DOX treatment [76,77], with the consequent impairment of mitochondrial oxidative phosphorylation, energy deficit, and oxidative stress. mtDNA damage was also observed in human autoptic myocardial samples of patients treated with DOX [78]. Ferreira and colleagues showed that DOX treatment in rats decreases global DNA methylation and alters the transcriptional levels of several mitochondrial genes encoded by both nuclear and mitochondrial genomes. The latter includes subunits of electron transport chain complexes I, III, and IV and genes involved in mitochondrial biogenesis [79].

### 3.2. Mitochondrial Dynamics

Mitochondrial biogenesis is not the only mechanism required for mitochondrial homeostasis. Recent evidence suggests that mitochondrial dynamics play a central role in cardiac homeostasis both in unstressed and stressed conditions. Mitochondria dynamics include fission, which consists of the fragmentation of irreversibly damaged mitochondria into spheroids and fragmented mitochondria, and fusion, which occurs in the presence of reversible damaged mitochondria and leads to the formation of elongated organelles [80]. Mitochondrial dynamics are orchestrated by different players, among which the best-characterized ones include dynamin-related protein 1 (DRP-1) and Mitochondrial fission factor (MFF) for fission and Optic atrophy 1 (OPA-1) and Mitofusin 1-2 (Mfn1-2) for fusion. A correct balance between fusion and fission is essential for cell survival, mitochondrial injury management, and stress response after DOX treatment. Melatonin and metformin preserve mitochondrial dynamics and structure, reduce apoptosis, and increase physiological energy metabolism and myocardial contractility in Wistar rats receiving chronic DOX treatment [81]. Pharmacological inhibition of excessive mitochondrial fission and mitophagy (i.e., mitochondrial-specific macro-autophagy) by liensinine administration also rescues apoptosis and cardiac dysfunction in mice acutely treated with 15 mg/kg DOX [82]. LCZ696, a novel combined angiotensin-II receptor and neprilysin inhibitor, similarly reduces DOX-induced DRP-1 activation, excessive mitochondrial removal, and reduction in cardiac contractility in mice receiving chronic DOX administration [83].

Besides fission, overexpression of essential mitochondrial fusion protein MFN-2 was found to rescue cardiac dysfunction in mice with chronic DOX treatment, an effect abolished by lentiviral-mediated DRP-1 overexpression. DOX-treated mice show a shift toward fission, with a Forkhead box protein O1 (FoxO1)-dependent transcriptional downregulation of Mfn2, increased ROS production, cardiomyocyte metabolic switch, and induction of apoptosis [84].

Overall, these studies show that DOX promotes mitochondrial fission to remove heavily damaged portions of the mitochondrial network, eventually leading to programmed cell death. Counteracting this stress response by mitigating fission activation or boosting fusion seems to be an effective approach to reducing cardiac derangements occurring in response to DOX treatment (Figure 3).

## 4. DOX Effects on Autophagy

Autophagy is a homeostatic and stress response cellular mechanism for the survival and functioning of mammalian cells, including terminally differentiated cell types, such as cardiomyocytes and neurons [85]. Compelling evidence suggests that defects in autophagy are mechanistically linked to cardiovascular pathologies, including heart failure, ischemia-reperfusion injury, and metabolic and genetic cardiomyopathies. Boosting autophagy with different strategies, such as caloric restriction, intermittent fasting, or pharmacological activators, reduced stress-induced myocardial injury [86]. Previous work demonstrated that both acute and chronic DOX-induced cardiomyopathies are associated with an accumulation in the mouse heart of Microtubule-associated protein 1A/1B-light chain 3b (Lc3b), a key marker of autophagosomes, as a result of the inhibition of the late phase of autophagic flux, i.e., the fusion of the autophagosome with lysosome and lysosomal vesicle degradation [87].

Autophagic flux inhibition may impair the degradation of damaged and dysfunctioning organelles leading to oxidative stress, mitochondrial dysfunction, and cell death. At the same time, flux inhibition may lead to exaggerated autophagosome accumulation, which may cause autosis, an autophagy-dependent form of cell death [88]. In support of this notion, genetic inhibition of autophagy initiation by heterozygous systemic Beclin1 gene deletion was found to alleviate DOX-induced cardiac dysfunction. This beneficial effect appeared to be associated with flux reactivation in mice treated with DOX because of the relief of the lysosomal system, which is otherwise exhausted by exaggerated autophagosome accumulation induced by DOX [89]. Conversely, mice with cardiomyocyte-specific Beclin1 overexpression display aggravated cardiac dysfunction in response to DOX treatment with respect to WT controls, associated with increased autophagosome accumulation. Of note, Beclin1 may also induce apoptosis by relieving Bcl-2-associated X protein (Bax) from the inhibitory effect of Bcl-2 and Bcl-xL through the sequestration of these proteins [90]. Thus, mice with Beclin1 overexpression may also have an increased rate of cardiomyocyte apoptosis, independently of autophagy.

Fasting induces autophagy by inhibiting the mechanistic target of rapamycin complex 1 (mTORC1) [91]. In this regard, it was previously shown that mice undergoing 48 h of starvation before DOX injections (two administrations, final dose 20 mg/kg) are protected from DOX-associated cardiac dysfunction and autophagy inhibition [92]. Starvation also rescues levels of AMPK and unc-51-like kinase 1 (ULK1), an autophagy-initiating kinase, which were found to be decreased in conventionally fed mice receiving DOX [92]. Conversely, mitochondrial aldehyde dehydrogenase (ALDH2) was found to protect the heart of mice treated with DOX for 6 days (final dose 15 mg/kg), and this effect was associated with apparent inhibition of autophagy induction [93]. However, this study did not assess autophagic flux, and mechanistic experiments aimed at dissecting the specific role of autophagy in the beneficial effects of ALDH2 were not carried out [94]. Rapamycin, a mTORC1 inhibitor, was also shown to rescue physiological myocardial structure and function in mice receiving DOX [95]. In the same study, the authors reported that systemic genetic deletion of the proinflammatory cytokine migration inhibitory factor (MIF) exacerbates DOX-induced cardiac dysfunction, impairs autophagolysosome formation, and increases mitochondrial dysfunction. These results suggest that MIF is protective against DOX-induced cardiotoxicity, likely through autophagolysosome formation.

Recently, it was also found that phosphoinositide 3-kinase γ (PI3Kγ) contributes to DOX-induced myocardial injury by reducing autophagy-dependent removal of damaged mitochondria. Mice with cardiac-specific overexpression of dominant-negative PI3Kγ have reduced mTOR signaling, preserved mitophagy after DOX, together with unaltered myocardial structure, increased cardiomyocyte survival, and preserved cardiac function. These effects are abolished by either pharmacological or genetic autophagy inhibition [96]. Interestingly, the cardiac effects of mTOR signaling in response to acute DOX treatment might differ from the detrimental effects observed in response to the chronic treatment. Previous work showed that sustained mTOR activation, either by cardiac-specific overexpression of constitutively active mTOR or by overexpression of a p53 inhibitor (MHC-CB7), preserves myocardial mass and function in response to acute anthracycline treatment [97]. On the other hand, p53 limits DCM in the long term [98]. Future studies are warranted to elucidate whether autophagy has different effects on acute vs. chronic DOX cardiotoxicity and the specific effects of mTORC1 vs. mTORC2.

Recent improvements in our capacity to dissect mitochondrial turnover led to an increased understanding of mitophagy [99]. However, only a minority of these studies were conducted on animal models. It was found that systemic loss of the Rubicon gene, a late-stage autophagy inhibitor, recovers Parkin-mediated mitophagy, mitochondrial dynamics, and myocardial structural integrity in mouse models of acute DOX administration [100]. Similarly, Sprague–Dawley rats receiving chronic DOX administration have reduced Parkin-dependent mitophagy and developed DCM. Adenoviral overexpression of Sestrin-2, which was found reduced after DOX treatment, preserves cardiac function and increases mitophagy in vivo by promoting Parkin localization on damaged mitochondria [101]. Administration of oseltamivir, a neuraminidase (NEU-1) inhibitor, suppresses DRP-1-dependent mitochondrial fission and PTEN-induced kinase 1 (PINK-1)/Parkin-induced mitophagy, thus protecting the heart from the development of contractile dysfunction, mitochondrial derangements and cell death in response to chronic DOX treatment [102].

Together these studies demonstrate that autophagy is inhibited in DCM and that recovery of autophagic flux and pre-treatment with autophagy inducers protect the heart from the development of structural and functional abnormalities, likely by promoting damaged mitochondrial removal (Figure 3). This effect should be achieved by limiting excessive mitochondrial fission. Another therapeutic strategy to reduce DCM should trim autophagy initiation, as it occurs in Beclin 1 KO mice. This strategy relieves DOX-induced autophagosomal accumulation due to compromised lysosomal function. In fact, minor autophagosome formation grants more time to dispose of cytoplasmic cargoes. Nevertheless, considering that Beclin1 is involved in many different pathways, including apoptosis, an advisable strategy may also be based on a Beclin1-independent activation of autophagy.

## 5. DOX Effects in Non-Myocytes

In addition to cardiac cells, the effects of DOX were also studied in non-myocyte cells, although the molecular mechanisms were less characterized. Zhan et al. demonstrated that mice with cardiac fibroblast-specific deletion of Ataxia telangiectasia mutated (ATM) kinase show reduced cardiotoxicity in response to DOX. A similar effect was also observed with the pharmacological inhibition of ATM, which was associated with reduced release of FAS-L by fibroblasts and decreased cardiomyocyte apoptosis [103]. In addition, p53 deletion in cultured cardiac fibroblasts improves mitochondrial dysfunction and reduces apoptosis in response to DOX [104], but, to date, this evidence was not corroborated by in vivo studies. The contributory role of CPCs and ECs in DOX-induced cardiotoxicity was also reported. The impairment of vascular development was observed, along with a reduced number of CPCs, in a model of late-onset DOX-induced cardiotoxicity [105]. In fact, DOX induces apoptosis in endothelial cells and impairs endothelial-dependent relaxation [106]. Microvessel density is also impaired by DOX, in association with increased miR-320a expression and decreased level of VEGF-A [107]. Future studies should test the mitochondria-specific effects of DOX in these cell types.

## 6. Other Anthracyclines and Cardiotoxicity

In addition to DOX, other anthracyclines (i.e., daunorubicin, epirubicin) were reported to induce cardiotoxicity. Daunorubicin (DAN) was the first identified anthracycline to treat leukemia. However, several pieces of evidence demonstrated its severe adverse cardiac effects both in preclinical models and in humans. A meta-analysis revealed that daunorubicin is less cardiotoxic among childhood cancer survivors compared to DOX [108]. At the molecular level, DAN was found to reduce the expression of MnSOD, impairing ROS scavenging [109]. Epirubicin (EPI) induced cumulative dose-related cardiotoxicity and CHF in humans, but it was considered safer than DOX. However, EPI also shows a worse therapeutic effect in cancer patients compared to DOX [110]. Salvatorelli et al. demonstrated that EPI generates less ROS compared to DOX in cardiac ex vivo samples and in H9c2 cells due to its limited localization to mitochondrial one-electron quinone reductases [111]. However, Toldo et al. also reported in vivo and in vitro that EPI and DOX have similar cardiotoxicity [112]. A different study showed that EPI administration induces severe cardiotoxicity in vivo, which was associated with the upregulation of death receptors and apoptosis [113]. Overall, these data suggest that future studies are still necessary to improve our understanding of anthracycline toxicity and develop effective cardioprotective strategies that may be used with different chemotherapeutic agents.

## 7. Perspectives

DCM remains a major issue for public health, considering that patients suffering from cardiomyopathy might develop various degrees of disability and undergo recurrent hospitalizations. Much effort is being made for the targeted delivery of both chemotherapeutic agents and cardioprotective drugs to neoplastic cells and cardiomyocytes, respectively. At state of the art, DOX packaging in targeted nanovesicles is one of the most promising pharmacokinetic approaches to mitigate non-specific cytotoxicity [114,115]. In this regard, encapsulating DOX in liposomes was reported to reduce the toxicity in cardiac cells in vitro [116]. A meta-analysis revealed that liposomal DOX-based chemotherapy reduces the risk of cardiotoxicity in breast cancer patients compared to patients undergoing conventional chemotherapy [117]. In the same line of evidence, exosome-based nanoparticles engineered with DOX were reported to show less cardiac toxicity [118,119,120] without reducing their antineoplastic activity.

From a pharmacological point of view, studies focusing on ROS scavenging (e.g., Coenzyme Q10, L-carnitine, acetylcysteine, vitamins C and E) provided disappointing results in the clinical translation [121], and dexrazoxane is the only option that gathers a large consensus [17]. Mito-TEMPO, a mitochondria-targeted antioxidant, was reported to reduce mitochondrial ROS in mice [38,122]. Natural activators of autophagy, such as trehalose and spermidine, were reported to exert beneficial effects in preclinical models of cardiovascular diseases [123]. It may be interesting to test their effects on the prevention of DCM. Targeting ferroptosis is another interesting approach to reducing DOX-induced cardiotoxicity. Ferrostatin-1, a widely used ferroptosis inhibitor, was found to reduce mitochondrial dysfunction and DCM in mice [124]. Lifestyle interventions, such as exercise, were also reported to be a preventive strategy for reducing DCM. Encouraging evidence in clinical studies showed that aerobic and resistance exercises improve systolic function and cardiorespiratory fitness in patients undergoing anthracycline treatment [125]. Innovative approaches targeting mitochondrial derangements may be developed in the near future. Since autophagy inhibition is a novel mechanism recently associated with DCM, it would be interesting to verify in the following years whether the administration of pharmacological activators of autophagy is beneficial and may represent future candidates for new cardioprotective approaches [126].

## Figures and Tables

**Figure 1 cells-11-02000-f001:**
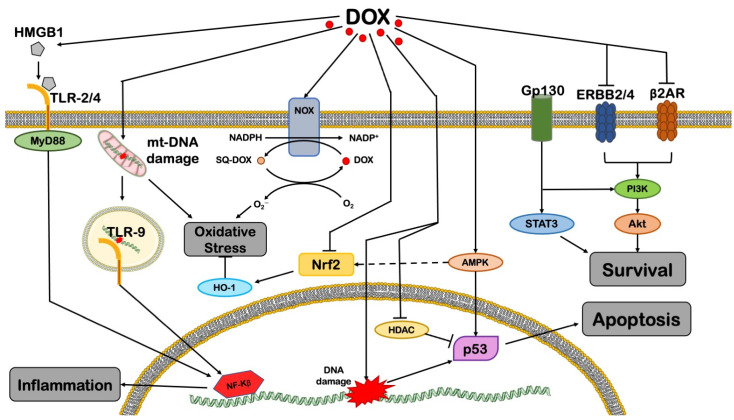
An overview of DOX-induced cytotoxicity in cardiomyocytes. The image reports the main mechanisms mediating the harmful effects of DOX. See text for further details. AMPK = AMP-activated protein kinase; AR = adrenergic receptor; Gp130 = glycoprotein 130; HDAC = histone deacetylase; HMGB1 = high-mobility group box 1; HO-1 = heme oxygenase 1; mtDNA = mitochondrial DNA; MyD88 = myeloid differentiation primary response 88; NADP = nicotinamide adenine dinucleotide phosphate; NF-kB = nuclear factor kappa-light-chain-enhancer of activated B cells; NOX = NADPH oxidase; NRF2 = nuclear respiratory factor 2; PI3Kγ = phosphoinositide 3-kinase γ; sq-DOX = doxorubicin (semiquinone form); STAT3 = signal transducer and activator of transcription 3; TLR = Toll-like receptor. This illustration includes elements from Servier Medical Art.

**Figure 2 cells-11-02000-f002:**
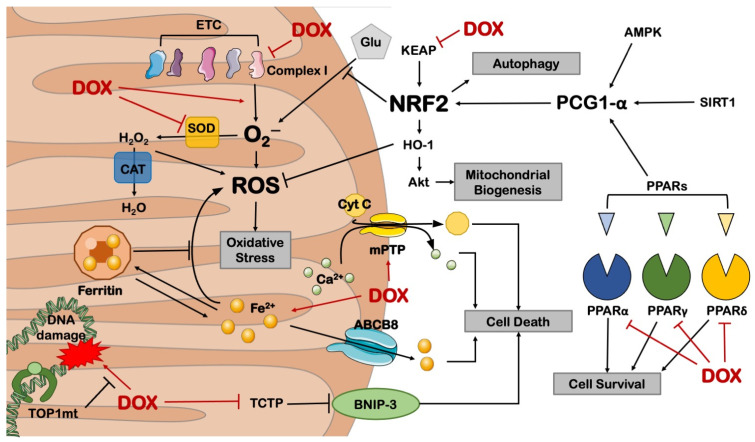
Mitochondrial derangements in DOX-induced cardiotoxicity. ABCB8 = ATP-binding cassette sub-family B member 8; AMPK = AMP-activated protein kinase; BNIP-3 = BCL2 19 kD protein-interacting protein 3; CAT = catalase; Cyt C = cytochrome C; ETC = electron transport chain; Glu = glucose; HO-1 = heme oxygenase 1; KEAP = Kelch-like ECH-associated protein; mPTP = mitochondrial permeability transition pore; NRF2 = nuclear respiratory factor 2; PGC-1α = PPARγ coactivator 1 alpha; PPAR = peroxisome proliferator-activated receptor; ROS = reactive oxygen species; SIRT1 = sirtuin 1; TCTP = translationally controlled tumor protein; TOP1mt = mitochondrial topoisomerase 1. This illustration includes elements from Servier Medical Art.

**Figure 3 cells-11-02000-f003:**
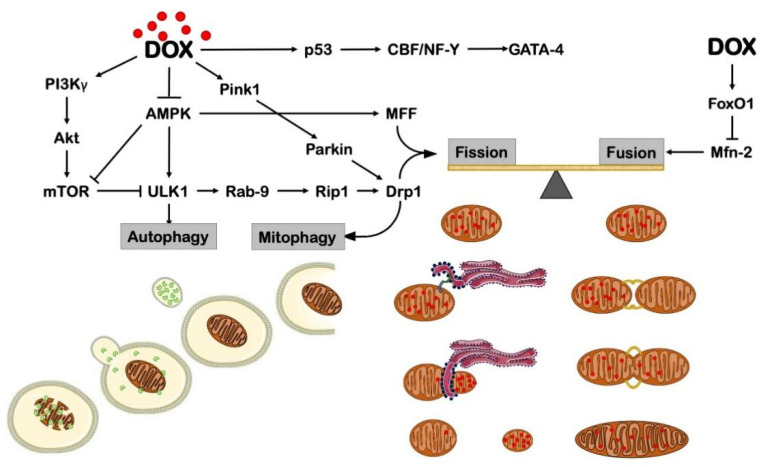
DOX impairs mitochondrial dynamics and mitophagy through different pathways. AMPK = AMP-activated protein kinase; CBF/NF-Y = CCAAT-binding factor/nuclear factor-Y; DRP1 = dynamin-related protein 1; FoxO1 = Forkhead box protein O1; GATA4 = GATA-binding protein 4; Mfn-2 = mitofusin 2; mTOR = mechanistic target of rapamycin complex 1; PI3Kγ = phosphoinositide 3-kinase γ; PINK1 = PTEN-induced kinase 1; RAB-9 = Ras-related protein Rab-9; RIP1 = receptor-interacting protein 1; ULK1 = unc-51-like kinase 1. This illustration includes elements from Servier Medical Art.

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
