# Peer review of "Mitochondria and Doxorubicin-Induced Cardiomyopathy: A Complex Interplay"

_cells, 2022, doi:10.3390/cells11132000_

Round 1
Reviewer 1 Report
This review article from Schirone and colleagues provides an overview of Doxorubicin-induced cardiomyopathy, with a focus on mitochondrial injury and dysfunction and its contribution to pathology. The manuscript is clear, logically structured, and well-written. The basic signaling mechanisms that regulate mitochondrial-mediated cell death, metabolic derangements, and altered mitochondrial dynamics and mitophagy are discussed in sufficient detail, albeit a lot of information is presented. To this end, the Figures are a helpful guide to the reader. The authors also provide disease background, including epidemiology and physiology, that informs and orients the audience to the clinical implications of these studies. Finally, future perspectives are provided in relation to potential therapeutic interventions. Overall, this review is excellent, and I have only a few suggestions for further improvement.
1. In section 4, the authors describe how altered autophagy/mitophagy contributes to DOX-induced cardiomyopathy and dysfunction. The role of Beclin 1 is discussed. Compared to other interventions aimed at modulating the induction of autophagy, results from studies testing Beclin 1 and its potential therapeutic manipulation appear to be less straightforward (e.g. Beclin 1 also interferes with apoptosis). Perhaps the nuance surrounding Beclin 1 interventions and its role in DOX-mediated DCM could be discussed in greater detail, e.g. in the autophagy concluding remarks.
2. Understandably, the authors focus on the effects of DOX in cardiomyocytes and the mitochondria therein. If anything related to the mitochondrial impact of DOX specifically in non-myocytes is known, perhaps that could be mentioned.
3. The final perspectives section could be expanded to include more specifics about how particular pathways might be targeted, and how this may lead to enhanced cardioprotection during DOX exposure. More details about studies using nanoparticles to package and deliver DOX could also be discussed here. This is exciting work that addresses a significant clinical need, and readers will be quite interested to learn more.
4. The authors should consider moving Figure 3 so that it is referenced earlier in the paper when discussing cell death and metabolic dysfunction (in section 2). This may be a better fit and improve overall readability.
5. Minor comment, there are some spelling mistakes, e.g. “cardiomyocytic” (line 113), “peripheric tissues” (line 186) that need correction.
6. Minor comment, there appears to be a typesetting issue on line 134.
Author Response
This review article from Schirone and colleagues provides an overview of Doxorubicin-induced cardiomyopathy, with a focus on mitochondrial injury and dysfunction and its contribution to pathology. The manuscript is clear, logically structured, and well-written. The basic signaling mechanisms that regulate mitochondrial-mediated cell death, metabolic derangements, and altered mitochondrial dynamics and mitophagy are discussed in sufficient detail, albeit a lot of information is presented. To this end, the Figures are a helpful guide to the reader. The authors also provide disease background, including epidemiology and physiology, that informs and orients the audience to the clinical implications of these studies. Finally, future perspectives are provided in relation to potential therapeutic interventions. Overall, this review is excellent, and I have only a few suggestions for further improvement.
We are very grateful to the Reviewer for the positive evaluation of our Review. We also thank for his/her comments. Please note that all relevant changes made in the text are highlighted in red font.
- In section 4, the authors describe how altered autophagy/mitophagy contributes to DOX-induced cardiomyopathy and dysfunction. The role of Beclin 1 is discussed. Compared to other interventions aimed at modulating the induction of autophagy, results from studies testing Beclin 1 and its potential therapeutic manipulation appear to be less straightforward (e.g. Beclin 1 also interferes with apoptosis). Perhaps the nuance surrounding Beclin 1 interventions and its role in DOX-mediated DCM could be discussed in greater detail, e.g. in the autophagy concluding remarks.
Thank you for this comment. As suggested, we added more details about the therapeutic interventions aimed at modulating autophagy, with a particular focus on the role of Beclin 1 (please see section 4, last paragraph).
- Understandably, the authors focus on the effects of DOX in cardiomyocytes and the mitochondria therein. If anything related to the mitochondrial impact of DOX specifically in non-myocytes is known, perhaps that could be mentioned.
The Reviewer raised an important aspect about the effects of DOX in non-myocytes. We added a new section about the effects of DOX in non-myocytes (please see section 5). Thank you.
- The final perspectives section could be expanded to include more specifics about how particular pathways might be targeted, and how this may lead to enhanced cardioprotection during DOX exposure. More details about studies using nanoparticles to package and deliver DOX could also be discussed here. This is exciting work that addresses a significant clinical need, and readers will be quite interested to learn more.
Thank you for this suggestion. We improved the final perspectives section. We also discussed studies regarding the use of nanoparticles to reduce the cardiac toxicity of DOX in more detail (please see section 7).
- The authors should consider moving Figure 3 so that it is referenced earlier in the paper when discussing cell death and metabolic dysfunction (in section 2). This may be a better fit and improve overall readability.
As suggested, we moved figure 3 after section 2. Thank you
- Minor comment, there are some spelling mistakes, e.g. “cardiomyocytic” (line 113), “peripheric tissues” (line 186) that need correction.
- Minor comment, there appears to be a typesetting issue on line 134.
We corrected all typos throughout the manuscript.
Reviewer 2 Report
Doxorubicin is a widely used chemotherapy drug for treatment of cancer patients of all ages, however acute and chronic cardiotoxicity in the cancer patient is the problem associated with its use. In this review article, the authors discuss the effects of doxorubicin treatment on mitochondria. structural and functional changes, biogenesis and clearance of mitochondria in cardiac dysfunction.
Comments:
This review article is summary of the effects of doxorubicin treatment on structural and functional changes, biogenesis and clearance of mitochondria and cardiac dysfunction. However, some important aspects are missing and should be included to provide a better and complete picture.
1.Male and female patients may respond differently to Doxorubicin treatment. The authors should include the studies that focused on Doxorubicin cardiotoxicity and sex related differences.
2. Age related differences in response to Doxorubicin should be included.
3 Since mitochondria are the focus of this review, authors should describe the changes in mitochondrial DNA encoded vs nuclear DNA encoded mitochondrial genes that change in response to Doxorubicin.
4.Authors should include a section about other anthracyclines and chemotherapy drugs and their effects on mitochondria.
5 Doxorubicin effect on cardiomyocytes may differ from how it might influence fibroblast and other cell types present in the heart . The authors should discuss that as well.
Author Response
Doxorubicin is a widely used chemotherapy drug for treatment of cancer patients of all ages, however acute and chronic cardiotoxicity in the cancer patient is the problem associated with its use. In this review article, the authors discuss the effects of doxorubicin treatment on mitochondria. structural and functional changes, biogenesis and clearance of mitochondria in cardiac dysfunction.
Comments:
This review article is summary of the effects of doxorubicin treatment on structural and functional changes, biogenesis and clearance of mitochondria and cardiac dysfunction. However, some important aspects are missing and should be included to provide a better and complete picture.
We are very grateful with the Reviewer for his/her careful evaluation of our Review. We took into great consideration all the criticisms. Please note that all relevant changes made in the text are highlighted in red font.
1.Male and female patients may respond differently to Doxorubicin treatment. The authors should include the studies that focused on Doxorubicin cardiotoxicity and sex related differences.
The reviewer raises an important aspect about risk factors concurring to cause the adverse cardiac effects of DOX. As suggested, we discussed sex-related differences in Section 1.
- Age related differences in response to Doxorubicin should be included.
We thank the Reviewer for this comment. We briefly discussed age-related differences in Section 1.
3 Since mitochondria are the focus of this review, authors should describe the changes in mitochondrial DNA encoded vs nuclear DNA encoded mitochondrial genes that change in response to Doxorubicin.
We totally agree with the Reviewer’s concern. Accordingly, we described the effects of DOX on mitochondrial DNA vs nuclear DNA encoded mitochondrial genes. Please see section 3.1, last paragraph.
4.Authors should include a section about other anthracyclines and chemotherapy drugs and their effects on mitochondria.
The effects of other anthracyclines on mitochondria and cardiac function are briefly discussed in Section 6.
5 Doxorubicin effect on cardiomyocytes may differ from how it might influence fibroblast and other cell types present in the heart. The authors should discuss that as well.
This is an interesting point. We added a new section that includes all relevant studies about the effects of DOX on cardiac fibroblast and other non-myocyte cells. Thank you